# Cancer Stem Cells and Radioresistance: DNA Repair and Beyond

**DOI:** 10.3390/cancers11060862

**Published:** 2019-06-21

**Authors:** Alexander Schulz, Felix Meyer, Anna Dubrovska, Kerstin Borgmann

**Affiliations:** 1OncoRay–National Center for Radiation Research in Oncology, Faculty of Medicine and University Hospital Carl Gustav Carus, Technische Universität Dresden, 01307 Dresden, Germany; Alexander.Schulz@uniklinikum-dresden.de; 2Laboratory of Radiobiology & Experimental Radiooncology, Department of Radiotherapy and Radiooncology, Center of Oncology, University Medical Center Hamburg-Eppendorf, 20251 Hamburg, Germany; fe.meyer@uke.de; 3Helmholtz-Zentrum Dresden–Rossendorf, Institute of Radiooncology–OncoRay, 01328 Dresden, Germany; 4German Cancer Consortium (DKTK), Partner Site Dresden, 01307 Dresden, Germany; 5German Cancer Research Center (DKFZ), 69120 Heidelberg, Germany

**Keywords:** cancer stem cells, DNA repair, radioresistance, 5Rs of radiation biology

## Abstract

The current preclinical and clinical findings demonstrate that, in addition to the conventional clinical and pathological indicators that have a prognostic value in radiation oncology, the number of cancer stem cells (CSCs) and their inherent radioresistance are important parameters for local control after radiotherapy. In this review, we discuss the molecular mechanisms of CSC radioresistance attributable to DNA repair mechanisms and the development of CSC-targeted therapies for tumor radiosensitization. We also discuss the current challenges in preclinical and translational CSC research including the high inter- and intratumoral heterogeneity, plasticity of CSCs, and microenvironment-stimulated tumor cell reprogramming.

## 1. Introduction

According to the global cancer statistics, cancer incidence and mortality are increasing worldwide with an estimated 29 million new cancer cases by 2030 [1,2,3,4], attributed to constant growth and aging of the human population [5]. This rapid growth of cancer rates suggests that improving diagnosis of cancer and integration of individualized treatment into the mainstream clinical practice are of utmost importance. As of now, about half of all cancer patients receive radiation therapy as a treatment modality during the history of disease [6]. The recent improvements in radiation technologies and delivery substantially increased efficiency and quality of treatment [6]. Nevertheless, one of the fundamental problems of radiation oncology is tumor resistance to radiation doses which cause an acceptable degree of normal tissue toxicity. Tumor radioresistance leads to loco-regional control failure and disease progression [7,8]. The curative potential of radiotherapy depends on its ability to cause a reproductive death of tumor cells via accumulation of non-repairable DNA lesions, thereby removing cancer cells from the clonogenic pool [9,10,11,12].

In the last few decades, studies in different fields of the biological sciences demonstrated that cancer cells are heterogeneous in their tumor-initiating properties which contribute to tumor growth and metastasis development [10,13,14,15]. Tumor cell transplantation experiments, first in isogenic murine models by Hewitt and Wilson and then in human xenograft tumor models, demonstrated that the therapeutic potential of radiotherapy is defined by the number of tumorigenic cells in these experimental tumors [9,12,16,17,18]. These findings are in agreement with clinical data for patients treated with radiotherapy; smaller tumor size was associated with longer patient survival because larger tumors might contain higher absolute number of tumorigenic cells. Together with other clinical and pathological parameters, tumor size is used for radiation therapy patient stratification [19,20].

Identification of markers for the detection of tumor-initiating cells, first in leukemia by the group of John Dick (CD (cluster of differentiation) 34^+^/CD38^−^) and then in solid tumors including glioma (CD133^+^, aldehyde dehydrogenase, ALDH1^+^), breast (epithelial cell adhesion molecule, EpCAM^+^/CD44^high^/CD24^low^, ALDH1^+^), colorectal (CD133^+^, EpCAM^high^/CD44^+^, ALDH1^+^), head and neck squamous carcinoma (CD44^+^), and other cancers as discussed elsewhere [21], paved the way for isolating, enriching, and analyzing these tumorigenic cells in different tumor entities [15,21,22]. These findings indicated that tumor-initiating cells, also called cancer stem cells (CSCs) possess two fundamental properties that make them different from other tumor cells; they have unlimited capacity to self-renew (e.g., divide asymmetrically to produce an identical copy of itself and more differentiated progeny cells) and differentiate to all cell populations present in the original tumors. These properties make these cells a root of tumor growth and recurrence and, thus, an important marker for tumor diagnosis, prognosis, and treatment, as well as critical targets for cancer therapy. Strong evidence is emerging to support the dynamic nature of tumor stemness which can be influenced by genetic alterations, epigenetic reprogramming and the tumor microenvironment [13,23]. Although tumor cell heterogeneity displays a much higher level of complexity than can be explained by the hierarchical model, CSC populations remain critical targets and biomarkers for cancer treatments.

In this review, we discuss the preclinical investigation of the CSC role in tumor radioresistance, clinical evidence for CSCs as potential prognosticators and therapeutic targets in radiation oncology, and potential challenges for their translation into clinical practice.

## 2. Cancer Stem Cells and 5Rs of Radiation Biology

Radiation therapy is one of the key anti-cancer treatment options along with surgery and chemotherapy [6]. However, a high inter- and intratumoral variability in the phenotypical and functional properties of cancer cells remains the major obstacle to improving cancer survival rates, which is especially true for patients with locally advanced tumors [15,24], and reliable biomarkers for patient stratification are of high demand [25]. As of today, only few biological parameters are suggested as potential biomarkers for radiation oncology including tumor size, hypoxia, or positivity for human papilloma virus (HPV) for head and neck squamous cell carcinoma (HNSCC) [11]. The biology-based patient stratification aims to select potential responders and non-responders to increase the probability of cancer cure by radiation therapy. It is based on the radiobiological concept of the “5Rs” which are repair, redistribution, repopulation, reoxygenation, and intrinsic radioresistance [26,27].

The curative potential of radiotherapy depends on the scale and quality of DNA damage in the exposed tumor tissues. Ionizing radiation induces different types of DNA damage with DNA double-strand breaks (DSBs) as major lethal DNA lesions. If the levels of radiation-induced DSBs exceed the DNA repair capacity of tumor cells, this might result in cell cycle arrest, tumor cell senescence, and death. The cell fate decision following radiation exposure depends on the amount of critical DNA damage. A low level of DNA lesions triggers DNA repair mechanisms and DNA damage checkpoints, which arrest cell cycle progression in the presence of DNA damage and allow cells to repair DNA before returning to the proliferative pool [28]. However, if the amount of DNA damage is unrepairable, the cells activate death programs [28,29]. The first “R” refers to DNA repair as one of the key determinants of tumor cell survival after radiation therapy. The cellular response to DNA damage is a complex process including activation of the DNA damage response pathways and DNA repair mechanisms as reviewed elsewhere [28,30]. In addition to direct DNA ionization, radiation-induced DNA damage is caused by highly chemically reactive species produced by radiolysis of water and by disruption of normal mitochondrial functions [30]. These chemically active molecules, called reactive oxygen species (ROS) and reactive nitrogen species (RNS), induce damage of DNA, protein oxidation, and lipid peroxidation, and might cause cell senescence and death [31]. These chemical species are regular by-products of cell metabolism and important second messengers [31,32]. Under physiological conditions, the intracellular concentration of ROS is maintained by the scavenging system which includes, e.g., glutathione, thioredoxin, and enzymatic proteins thioredoxin reductase, dismutase, peroxidase, and catalase, all of which might also shield the cells against radiation-induced oxidative stress [33].

Accumulating preclinical evidence indicates that many of these protective mechanisms are activated in CSC populations possibly resulting in treatment resistance (Figure 1). These findings also suggest that signaling pathways that control DNA integrity and repair in CSCs may serve as a promising target in cancer therapy, as discussed in the next sections of this review.

To repair DSBs, cells employ two major mechanisms: the more error-prone non-homologous end joining (NHEJ) and the more accurate homologous recombination (HR). In addition, tumor cells also have two extremely error-prone DSB back-up repair mechanisms for both NHEJ and HR, the alt-EJ and single-strand annealing (SSA) [34]. In mammalian cells, HR occurs only in the late synthesis (S) phase and less in the gap 2 (G2) phase of the cell cycle when the DNA template on the sister chromatid is available for recombination, whereas NHEJ is active throughout the entire cell cycle with highest efficiency during the G2/mitosis (M) stage and is predominant in G0, G1, and early S phases [35,36,37,38,39].

An activation of these different DNA repair mechanisms at specific phases of the cell cycle results in differences in radiosensitivity throughout the cell cycle, with increased radioresistance in the late S phase and increased radioresponsiveness in G2 and M phases [40]. Increased cell radioresistance in the S phase was attributed to an increased level of DNA replication enabling the HR process [40]. Resistance caused by HR-mediated repair is further enhanced by the presence of all available DNA repair pathways, including those that go beyond the repair of DSBs [39,41]. Therefore, the second “R” refers to redistribution of tumor cells in the cell cycle during radiotherapy to the more radiosensitive cell states. Similar to normal stem cells, some CSC populations are slow-proliferating or quiescent cells [42]. A comparative analysis of DNA repair in the proliferating and quiescent tissues showed that quiescent cells lack DNA repair efficiency and display sustained accumulation of DSBs that could cause an activation of p53 signaling and apoptosis [43,44]. In contrast to the normal tissues and tumor bulk, CSCs have attenuated activation of p53 after radiation-induced DNA damage and, as result, impaired cell cycle arrest and apoptosis that, in the long run, might lead to accumulation of DNA mutations [15,44]. This accumulation of mutation burden over time increases intratumoral heterogeneity and leads to tumor evolution and disease progression [15]. The CSC evolution during tumor development and treatment is associated with activation of different pro-survival pathways which cause intrinsic radioresistance of tumor cells such as epidermal growth factor receptor (EGFR), phosphatidylinositol 3-kinase (PI3K)/Akt/mammalian target of rapamycin (mTOR), wingless-type MMTV integration site family (WNT), Notch, and Hedgehog signaling [11,14,45], and different agents targeting these molecular pathways are currently in preclinical and clinical development.

In addition to the intrinsic mechanisms of radioresistance, the fate of tumor cells after radiotherapy depends on the plethora of cues coming from the tumor microenvironment. Tumors show growth comparable to that of rapidly proliferating tissues with a comparably high demand for oxygen and nutrients from the blood vessels. A number of recent studies demonstrated that tumors can make their own blood vessels and some CSC populations, e.g., in glioblastoma, are capable of differentiating into endothelial cells [46,47]. However, sometimes tumors outgrow their vasculature and develop regions with an inadequate nourishment and supply of oxygen. The hypoxic microenvironment might promote radiotherapy resistance by decreasing DSB burden mediated by oxygen-dependent free-radical production as discussed elsewhere [48,49]. Hypoxia induces significant changes in tumor metabolism due to the deficiency of nutrients, low concentration of oxygen, and deregulation of transporter proteins and metabolic enzymes [49]. An impaired mitochondrial respiration in tumor cells under hypoxia increases glucose uptake to cover high energetic demands of growing tumors. This biological feature is used in clinical practice for tumor imaging by positron emission tomography (PET) using radiolabeled glucose analog fluorine-18-2-fluoro-2-deoxy-d-glucose (^18^F-FDG) [50,51]. Sustained hypoxia can also trigger mechanisms important for the maintenance of CSCs such as hypoxia-inducible factor (HIF) signaling, autophagy, and epithelial–mesenchymal transition (EMT) which is associated with acquisition of the mesenchymal characteristics by tumor cells. Recent data suggest that HIF signaling is important for the regulation of CSC properties in glioblastoma, ovarian cancer, and breast cancer, and also contributes to the activation of autophagy and EMT [49,52,53,54,55]. Autophagy is a mechanism of cellular “self-eating” by which cells sequestrate and digest certain organelles and protein complexes in a process of clean-up of damaged structures and misfolded proteins [56]. Autophagy might decrease efficiency of radiotherapy by its contribution to CSC maintenance and reduction of ROS-associated DNA damage [56,57,58]. A growing body of pre-clinical data demonstrated that EMT-related signaling pathways contribute to tumor cell reprogramming into CSCs, metastatic tumor spread, and radioresistance [59,60,61,62]. Taken together, these findings suggest that increasing the content of molecular oxygen in the tumor, or tumor reoxygenation, might be an effective way to increase radiation-induced DNA damage and to inhibit the signaling pathways and mechanisms favorable for CSC survival.

Tumor recurrences arise after radiotherapy depending on the actively proliferating tumor progenitor cells induced by tumor reoxygenation [63]. Tumor cell repopulation in the course of treatment might cause therapy failure. In support of this, clinical studies demonstrated that shortening the overall treatment time for fast-proliferating tumors prevents tumor repopulation and contributes to improved local tumor control [64]. The unlimited self-renewal capacity of CSCs suggests their role in tumor repopulation between the radiotherapy fractions. The recent computational modeling of HNSCC growth demonstrated that tumor re-oxygenation during treatment can result in substantial increase in the probability of CSC symmetric division (up to 50% as compared to 2% before treatment) that yields an increase of the CSC population within the tumor up to 30–35% [65].

Hereby, the radiobiological concept of 5Rs has to be considered in context of the cancer stem cell tumor model to explain the effect of radiotherapy on tumor tissues. In the next chapters, we discuss the DNA repair and further mechanisms of CSC therapy resistance to identify them as potential targets for the specific elimination of CSC populations.

## 3. Molecular Mechanisms of CSC Radioresistance

### 3.1. DNA Repair Factors and Pathways Upregulated in CSCs

It was previously observed that CSCs, comparable to tissue stem cells, have altered DNA damage response and repair pathways (Figure 2). In tissue stem cells, the alteration of DNA repair pathways ensures error-free DNA preservation. In CSCs, on the other hand, this causes resistance to exogenously induced DNA damage that leads to the failure of tumor therapy [66,67]. After irradiation, resistance correlates with significantly less DNA damage in glioma and breast cancer stem cells [68,69,70]. This also confirms the generally accepted view that tissue stem cells guarantee the maintenance of genomic stability and CSCs ensure the survival of the entire tumor population.

Most of the investigations regarding radioresistance and DNA repair were performed in glioblastoma. Radioresistance is mediated by enhanced activation of the two serine threonine protein kinases ataxia–telangiectasia mutated (ATM) and ATM- and Rad3-Related (ATR) and their two downstream checkpoint kinases (Chk1 and Chk2) [68,69,82,86,87]. Both kinases regulate the complex DNA damage response by initiating cell-cycle checkpoint control and activating corresponding DNA repair pathways. CD133^+^ glioma stem cells show an increased Chk1-dependent checkpoint response [9,69,86]. An increased expression of *NBS1*, a component of the Mre11, Rad50, Nbs1 (MRN) complex [88] involved in DSB sensor activity, was also detected. Increased DNA damage response and repair gene expression were also observed in CD133^+^ CSCs in lung carcinoma cells, breast, breast carcinoma-inducing cells, pancreatic CSCs, and non-small-cell lung cancer [41,71].

Starting point for the enhanced DNA damage response appears to be a general adaptation to increased replication stress and increased oxidative damage already in the untreated state, which leads to increased activation of the DNA damage response even after irradiation [84,89]. The DNA damage response is then mediated either by induced DSBs for amplified activation of ATM or by increased replicative stress for amplified activation of ATR. This results in a lower number of DSBs [89,90].

With regard to DNA double-strand break repair pathways in CSCs, HR is of outstanding importance, whereas, for NHEJ, only ATM-mediated effects are observed [91,92,93]. HR is the preferred DSB repair pathway in the S phase and it is more strongly activated by ATR and ATM in cells tolerating replication stress [68,72]. RAD51 overexpression, the main HR DNA repair protein, is observed in glioma stem cells and decreases at the transition to progenitor cells [92]. Glioblastoma cell lines show an upregulation of HR genes, especially *RAD51*, *BRCA1*, and *BRCA2*, which leads to less DNA damage after irradiation [74]. Thus, most studies suggest that resistance of CSCs manifests during the S phase by HR, regulated by MRN and activated by ATM/ATR.

The importance of S-phase repair also appears for CSC resistance mechanisms after chemotherapy in ovarian cancer. Here, resistance is observed by enhanced activation of translesion synthesis mediated by polymerase eta [79], as well as in HNSCC, where overexpression of the Fanconi anemia DNA repair proteins was observed in ALDH1-positive tumor cells [81].

However, a number of studies showed no difference or even lower DNA damage response in CSCs [41,91,94,95,96]. These contradictory observations suggest that an improved DNA damage response may not be a common feature of CSCs. More obvious is that CSCs and non-CSCs are transient populations and that, in addition to inter-tumoral heterogeneity, intratumoral heterogeneity must also be considered in DNA damage reaction functionality [96]. This was previously observed in glioblastoma, where *RAD51* expression decreases from CSC to progenitor cells [92].

### 3.2. Factors Indirectly Influencing DNA Repair Capacity of CSCs

In CSCs, radioresistance determined by DNA repair capacity may also be attributed to lower indirectly induced ROS causing DNA damage and ROS-dependent apoptosis [14,70]. Breast CSCs express higher concentrations of ROS scavengers and neutralize radiation-induced ROS [89]. In addition to the known proteins with ROS scavenger function, the multifunctional protein apurine/apirimidine endonuclease/redox effector factor (Ape1/Ref-1) is also increasingly expressed in CSCs. Among other functions, Ape1/Ref-1 is part of the DNA repair complex base excision repair (BER), so that Ape1/Ref-1 can reduce both intracellular ROS and increase DNA repair [68]. Radioresistance in mesenchymal CSCs indirectly influencing DNA repair capacity could also be due to nicotinamide *N*-methyltransferase (NNMT) overexpression through depletion of the accessible amounts of nicotinamide, which is a known inhibitor of cellular DNA repair mechanisms [85].

An additional resistance mechanism only indirectly linked to improved DNA repair is the induction of CSC quiescence. By stopping the cells in the G0 phase, DNA damage can be eliminated before entering the S phase [44,97]. This shift in the cell cycle distribution may explain why NHEJ was considered important for increased radiation resistance in recent studies [68,76,98,99,100].

Another observed phenomenon indirectly linked to DNA repair mediated resistance is that genotoxic treatment, such as irradiation and chemotherapy, itself leads to the reprogramming of non-CSCs into CSCs [22,75,83,101,102,103,104,105]. In reprogrammed stem-like cells, either a faster repair mediated by *BRCA1* and *RAD51* after gemcitabine in pancreatic cancer [75] or a stronger activation of ATR/Chk1 in colon carcinoma after treatment with DNA interstrand-crosslinking (ICL) agents was shown [83]. Zhang and colleagues even went so far as to postulate a direct dependence of the DNA signaling cascade and stem-cell characteristics. They observed an ATM-mediated stabilization of zinc finger E-box binding homeobox 1 (ZEB1) leading to an enhanced Chk1-dependent DNA damage response in previously epithelial breast cells [104]. This direct dependence on stem cell character and HR or S-phase DNA repair was also observed for breast epithelial cells. Depletion of *BRCA1* and *FANCD2* led to reprogramming in breast epithelial cells to mesenchymal phenotype [105].

## 4. CSC Heterogeneity and Plasticity

Tumor tissues constitute a heterogeneous population of cancer cells. Among them are CSCs with distinct clinically relevant properties, such as tumor-initiating capacity, therapy resistance, dormancy, and increased metastatic potential. Different models were generated to describe this intratumoral heterogeneity. Clonal evolution is a nonhierarchical model characterized by acquisition of mutations that allow emergence and expansion of a dominant clone by a growth advantage that increases frequency of this clone over time. The classical CSC model is hierarchical and hypothesizes an asymmetric division of a CSC, resulting in a stable number of CSCs. Finally, strong experimental evidence is accumulating to support CSC plasticity; a conversion of a CSC into a non-CSC phenotype can be reversed as a result of genetic mutations, epigenetic alterations, or microenvironmental changes. All these cues not only impact the fundamental CSC properties such as their capacity to self-renew and to differentiate, but also affect the proliferative potential, therapy resistance, and metastatic capacity of CSCs and their progenies [13,23,106]. Because no single model can entirely explain the complexity and behavior of a tumor, it is likely that these mechanisms contribute to heterogeneity in parallel. Kreso and Dick combined these models to the united model of clonal evolution [13].

### 4.1. EMT and CSC Phenotype

Although the proportion of CSCs in a tumor is generally low, the CSC population is divergent itself due to acquisition of different mutational loads, epigenetic changes, or cellular plasticity. All of these factors may be influenced by environmental factors like hypoxia, release of growth factors and cytokines, or interaction of CSCs with stroma and extracellular matrix. In fact, even ionizing radiation (IR) itself is able to induce changes in CSCs. For example, IR is able to induce EMT and metastasis, all of which are features closely linked to a CSC phenotype [107,108,109,110]. Whether or not EMT is associated with CSCs is currently still heavily debated. However, a rising body of evidence supports the idea that EMT at least in part contributes to features of CSCs [111,112,113]. In line with this, major transcription factors of the EMT signaling cascade like Snail family transcriptional repressor (Snail), ZEB1, or Twist family BHLH transcription factor 1 (Twist1) were shown to promote stemness properties [114,115]. In this context, Snail not only plays a crucial role in IR-mediated activation of EMT, migration, and invasion [116], but it also confers resistance to radiotherapy in colorectal cancer cells [117]. ZEB1, on the other hand, represses microRNAs like miR-183, miR200c, and miR203, which are known to inhibit stemness. The repression of these microRNAs essentially leads to upregulation of stem-cell factors SRY-box 2 (Sox2) and Kruppel-like factor 4 (Klf4) [118]. Finally, Twist1 positively regulates BMI1 proto-oncogene (Bmi-1), thereby inducing EMT and stemness [119]. Notably, ZEB1 and Twist1 were recently identified as downstream targets of fibroblast growth factor receptor 1 (FGFR1)/forkhead box M1 (FOXM1) in glioblastoma, and their expression is highly associated with resistance to radiotherapy [120]. Moreover, purified breast CSCs were shown to be more radioresistant when treated with transforming growth factor beta 1(TGF-β1) compared to their parental counterparts [121]. It was shown that IR itself can contribute to enhanced TGF-β1 release involving transcription factor activator protein 1 (AP-1) [122]. Secreted TGF-β1 remains inactive upon binding as homodimer to the latent TGF-β binding protein, but can be activated by IR-induced ROS [123]. The active form of TGF-β1 promoted stemness in breast and lung cancer cells by upregulating stem cell factors octamer binding protein 4 (Oct4), Sox2, Nanog, and Klf4 [124,125]. A study on radioresistant esophageal squamous cell carcinoma cell lines confirmed that these cell lines gained EMT properties and stem quality, and that radioresistance was found to involve a chromatin-associated protein, high-mobility group box 1 protein (HMGB1), which promotes DNA damage repair by Wnt signaling [126].

IR not only affects TGF-β signaling but also signaling pathways that are important for stem cell maintenance and are frequently exploited by cancer cells such as the Wnt, Hedgehog, or Notch pathways. The fact that all of these pathways are involved in EMT further supports the link between EMT and stemness. Although both features seem to be connected, there are publications suggesting that extent to which EMT is triggered may eventually determine whether or not stemness is also affected. Beck and colleagues could show in mouse skin squamous cell carcinoma (SCC) cells that stemness is only induced by lower levels of Twist1 [127]. Similarly, the constitutively high expression of Snail repressed stemness features while successfully inducing EMT in bladder and prostate cancer [128]. A breast cancer xenograft model demonstrated that migrating cells that were subject to EMT switched back to an epithelial state when reaching their metastatic destination, indicating that they retained ability for reversible plasticity [129]. Taken together, since EMT is a dynamic process with many transitional states and CSCs also rely on epithelial characteristics for extravasation and seeding metastasis, there is a need for deeper understanding of these intermediate states. It is likely that preferred stage of CSCs is as adaptive as possible for escaping treatment. This may be achieved by maintaining itself rather in a transitional than determined epithelial or mesenchymal stage.

### 4.2. CSC Induction by IR and ROS

IR-induced EMT is also partially achieved by creating ROS [130] which, in turn upregulates Snail via a mechanism involving the extracellular signal-regulated kinase 1/2 (ERK1/2) and glycogen synthase kinase 3 (GSK3β) [131]. IR-induced ROS is either directly produced by water radiolysis or indirectly by mitochondrial damage or metabolic alterations [132]. It is generally well accepted that produced ROS molecules damage proteins, lipids, and DNA, further impairing DNA damage repair. Due to their high metabolism and extensive use of glycolysis for energy production, cancer cells usually harbor higher ROS levels than non-malignant cells. Given that CSCs are less proliferative and show features of quiescence, it is not surprising that they display lower levels of intracellular ROS. This observation is associated with elevated free-radical scavengers [89]. Thus, CSCs can neutralize ROS more efficiently. Because they initially contain lower amounts of ROS before irradiation, treatment will end up with lower levels of ROS per CSCs compared to surrounding non-CSCs. Together with their ability to neutralize ROS more efficiently, CSCs are, therefore, more resistant to radiation treatment [89]. This way, some CSCs are spared by radiation treatment. Transcription factors can influence levels of antioxidant proteins. For example, nuclear factor kappa-light-chain-enhancer of activated B cells (NF-kB) is essential for CSCs to withstand stress exerted by ROS [133]. The nuclear factor erythroid 2-related factor 2 (NRF2) is another transcription factor which is activated upon oxidative stress to bind to the antioxidant response element (ARE) of promoter regions resulting in cancer initiation and progression, as well as stemness characteristics [134,135]. NRF2 was also shown to contribute to radioresistance by decreasing ROS in lung squamous carcinoma [136]. ROS can lead to upregulation of proteins which are closely related to stem cells. For instance, it was shown in a lung cancer model that IR-induced ROS upregulates the C–X–C motif chemokine receptor type 4 (CXCR4) which then senses its ligand C-X-C motif chemokine ligand 12 (CXCL12) [137]. In summary, while IR and ROS trigger tumor-associated cell death, they also create a favorable environment for conversion of non-CSCs to a CSC-like phenotype. This was shown for some types of malignancies including glioblastoma and breast cancer, and represents a possible challenge to IR treatment [22,101,102,103].

### 4.3. Impact of Tumor Microenvironment (TME) on CSCs 

The TME was long underestimated for determining the fate of primary malignancies. This is in part due to the limitations of *in vitro* models to adequately address this question. The TME is composed of stroma adjacent to the tumor periphery, immune cells, cancer-associated fibroblasts (CAFs), vascular endothelial cells, and secreted molecules assembling the extracellular matrix (ECM), including growth factors and cytokines. A detailed outline of how the TME may be therapeutically targeted can be found in a comprehensive review by Junttila and de Sauvage [138]. IR can act as a TME modifier via different mechanisms, which include activation of CAFs (and release of growth factors), changes in protein expression, induction of hypoxia, or inflammation associated with infiltration of immune cells. CSCs prefer defined areas within the TME that provide optimal conditions for maintaining their CSC characteristics. The area in which CSCs reside is defined as the CSC niche.

### 4.4. The CSC Niche

Similar to normal stem cells, CSCs reside in specialized niches. The CSC niche represents a very distinct location within the TME that is able to host CSCs by maintaining optimal conditions for their phenotype. The most prominent types of CSC niches are the perivascular and the hypoxic niche.

The perivascular niche is localized in close proximity to blood vessels like the sinusoid and supplies CSCs with nutrients, growth factors, and cytokines; it was identified to play a role in glioblastoma and head and neck squamous carcinoma [139,140]. Glioblastoma multiforme (GBM) and other solid tumors may also create hypoxic niches upon growth which CSCs exploit for colonization. This environment can induce stem-cell factors like Oct4 and Nanog to aid in conversion of non-CSCs to CSCs [141,142]. Hypoxia leads to activation of hypoxia-inducible factor (HIF) and transcription of direct target genes. The result is an increased resistance to drugs and irradiation [49].

It is hypothesized that some niches may also promote entering CSCs into a quiescent/dormant state, which enables them to resist treatment but can be reversed upon potential triggering signals resulting in tumor relapse. Although data on resistance of dormant cancer cells to irradiation are very poor, there are numerous examples when this state is associated with increased chemotherapeutic resistance. In a colorectal cancer model, for example, dominant clones emerged from surviving dormant clones after oxaliplatin treatment [143]. A similar outcome was observed with quiescent CSCs in bladder cancer xenografts and mouse squamous cell carcinoma in response to chemotherapy [144,145]. Not only do slow-cycling CSCs survive treatment, they can also initiate repopulation as a direct result of the treatment stress response which is of high importance for controlling relapse. However, CSCs can stay dormant for a prolonged period of time, creating micrometastases up to years after treatment. A very interesting study on breast CSCs identified the perisinusoidal niche as the preferred location for homing of dormant CSCs before initiating bone metastases by performing *in vivo* intravital confocal microscopy. Mechanistically, the distinct roles for E-selectin binding and CXCR4 stimulation were uncovered. In detail, E-selectin was a crucial factor for entry of CSCs (CD44^+^CD24^−/low^) but not non-CSCs (CD44^+^CD24^+/high^) to bone, while CXCL12/CXCR4 signaling promoted anchorage of CSCs inhabiting that niche. Consequently, abrogating E-selectin binding significantly decreased homing of CSCs, and inhibition of CXCL12/CXCR4 interaction led to mobilization of these cells into the blood stream [146]. It remains speculative whether the quiescent/dormant state may also be responsible for escaping treatment from IR. Therefore, more research elucidating this question is strongly needed.

It is believed that most niches can protect CSCs from chemo- and radiotherapy. Therefore, targeting these niches represents a promising approach to eradicate CSCs. Just recently, Chiblak and colleagues successfully demonstrated in multiple glioma models that carbon irradiation is capable of eradicating both hypoxic and stem-cell like tumor cells, while providing an antiangiogenic but immunopermissive environment for improved treatment [147]. It is of importance to point out that the relationship between niche and CSC is bidirectional as both compartments can influence each other. This was previously demonstrated for glioblastoma where the hypoxic niche supports CSC phenotype, while the CSCs themselves can influence vascularization by shaping the endothelium [46,47,142,148].

In conclusion, niches represent a protective environment for CSCs and are important for CSC maintenance and survival. Hence, future therapeutic approaches should not only aim at specifically targeting CSCs but also find solutions for modifying their niche to become undesirable for CSCs.

### 4.5. Exosomes and microRNA 

Exosomes are 30–100-nm membrane vesicles that are released by the cell of origin for communication to target cells. They typically carry lipids, proteins, DNA, messenger RNA (mRNA), and miRNA as cargo which is released into the recipient cell. Exosomes produced by stem cells were shown to contain stemness-associated transcription factors Oct-4, Nanog, and Wnt family members for altering cellular plasticity of their respective target cells [149,150]. One interesting approach demonstrated that exosomes from irradiated mesenchymal stem cells increased irradiation efficacy and metastasis control [151]. On the other hand, exosomes isolated from irradiated HNSCC cells boosted the migratory phenotype of recipient cells, indicating that the cell origin and the environmental stress factors such as irradiation play an essential role in determining the composition of exosomes [152].

MicroRNAs are also a frequently found cargo of CSCs. A high-throughput screen of prostate CSCs revealed that miR-139 and miR-183 could alter the pre-metastatic niche [153]. Regarding radiotherapy, most data are restricted to prostate cancer. By employing a NanoString-based miRNA assay containing 800 miRNAs, miR-601 and miR-4516 were identified to predict biochemical failure after salvage IR following radical prostatectomy in 43 prostate cancer patients [154]. A screen of miRNA mimetics in prostate cancer cell lines revealed multiple miRNAs to sensitize cells to IR. Mechanistic follow-up studies demonstrated that one of them, miR-890, directly targeted three DNA repair genes simultaneously (mitotic arrest deficient 2 like 2 [*MAD2L2*], WEE1 G2 checkpoint kinase [*WEE1*], and XPC complex subunit [*XPC*]), thereby delaying DNA damage repair [155]. Another study on prostate cancer cell lines showed the radiosensitizing effect of miR-145, whose expression level could predict radiotherapy response in 30 prostate cancer patients and *in vivo* xenotransplantation assays. MiR-145 targets, among others, DNA repair genes *RAD51*, MCL1 apoptosis regulator (*MCL1*), F2R like thrombin or trypsin receptor 3 (*F2RL3*), and Poly(ADP-Ribose) polymerase 1 (*PARP1*) [156]. While the same miR-145 was previously shown to suppress migration and invasion in prostate cancer, the delivery of this inhibitory sequence by nanoparticles to glioblastoma cells significantly reduced stem-cell-like properties and resistance to radiochemotherapy [157,158]. *In vitro* analysis of irradiated LNCaP and C4-2 prostate cancer cell lines resulted in abolished expression of miR-521 as compared to their non-irradiated counterparts. Overexpression of an miR-521 mimic increased radiosensitization, whereas treating cells with an miR-521 inhibitor highly increased radioresistance, suggesting a potential role of this miRNA in resistance to IR therapy [159].

Taken together, while there was effort on elucidating the contribution of miRNA signaling in radiation research, the impact of exosomal communication for radiation resistance is not well studied. There is a need to explore exosomal cargo and mechanisms as they can potentially be used as biomarkers for earlier diagnosis or may carry information about tumor type, environmental conditions, or metastases to predict prognosis and treatment response in future medical applications.

## 5. Targeting of DNA Repair Mechanisms in CSCs

Assuming that DNA repair pathways are upregulated in CSCs, and that radiation or genotoxic substances themselves lead to increased CSC accumulation, amplified DNA repair pathways should be inhibited in parallel to radiochemotherapy. The most promising candidates are currently inhibitors that help to increase replicating stress in CSCs, such as ATM, ATR, Chk1, PARP1, and RAD51 inhibitors [69,82,84,86,87,160,161,162,163]. The inhibition of Chk1 reduced the CSC pool in non-small-cell lung cancer cells, suggesting that DNA repair inhibitors are a promising targeting strategy for CSCs [86]. Inhibition of PARP1 led to significant radiation sensitization by suppressing self-renewal, inhibiting growth, and increasing DNA damage in glioma stem cells [160]. Conversely, ALDH1+ ovarian cancer cells with platinum resistance were observed to have PARP1 upregulation. Inhibition of ALDH1 leads to increased replication stress [164]. In triple-negative breast cancer (TNBC), PARP1 resistance appeared to be mediated by RAD51. Here the RAD51 inhibition could lead to a clear sensitization [73]. Due to the inhibition of RAD51, a delay of the G2 arrest led to a clear radiation sensitization in glioblastoma [163]. The inhibition of ATM also selectively led to radiation sensitization in glioblastoma CSCs [82,87]. Further potential candidates such as WEE1, DNA-PKCS, Chk1 indirectly via mTOR, and Akt led to radiation sensitization in individual observations, but must be confirmed in further studies [76,80,99,165,166,167]. The use of PARP inhibitors in combination with temozolomide and radiotherapy is currently being tested in clinical trials [168,169].

Since multiple pathways for multiple DNA damages can be upregulated simultaneously in CSCs, as shown in glioma stem cells, a multi-targeting approach with combined inhibition of DNA repair and cell-cycle control mechanisms can be an even more promising strategy to overcome CSC resistance [82,84]. For example, chronic replication stress in glioma stem cells triggered by collision of replication forks with DNA–RNA hybrids led to a permanent activation of Discoidin domain receptor tyrosine kinase (DDR), which mediated radiation resistance. Here, the inhibition of ATR and PARP-1 led to radio-sensitization [84] (Table 1).

## 6. Conclusions

The fundamental properties of CSCs, such as ability to self-renew, to differentiate to other tumor cell populations, and to withstand cancer treatment, make them the root of cancer growth and relapse and an attractive target for anti-cancer treatment. CSCs can be protected from radiation-induced damage by the hypoxic microenvironment and protective niche, as well as by a number of intrinsic mechanisms. The intrinsic radioresistance observed in CSCs, among others, is due to an upregulated DNA damage reaction. Most studies indicate that resistance of CSCs during the S phase under DNA replication is manifested by HR, regulated by MRN, and activated by ATM/ATR. The activation of the two serine/threonine protein kinases ATM and ATR and their two downstream kinases Chk1 and Chk2 played a major role, followed by the important role of the DNA repair mechanism, homologous recombination. The starting point for an improved DNA damage reaction appears to be a general adaptation to increased replication stress and increased oxidative damage already in the untreated state, which leads to an increased activation of the DNA damage reaction even after irradiation. Since the DNA repair pathways upregulated in CSCs are part of the radiation resistance observed in CSCs, inhibitors of the corresponding DNA repair proteins, such as ATM, ATR, Chk1, PARP1, and RAD51, alone or in combination, should be used in parallel to combined radiochemotherapy. This would lead to increased replication stress especially in CSCs, which could be an even more promising strategy to overcome CSC resistance in a possible multi-targeting approach with combined inhibition of DNA repair and cell-cycle control mechanisms. This approach in combination with epigenetic drugs could prevent the plasticity of cancer cells, the adaptive construction of chromatin, and the development of therapy-resistant mechanisms.

## Figures and Tables

**Figure 1 cancers-11-00862-f001:**
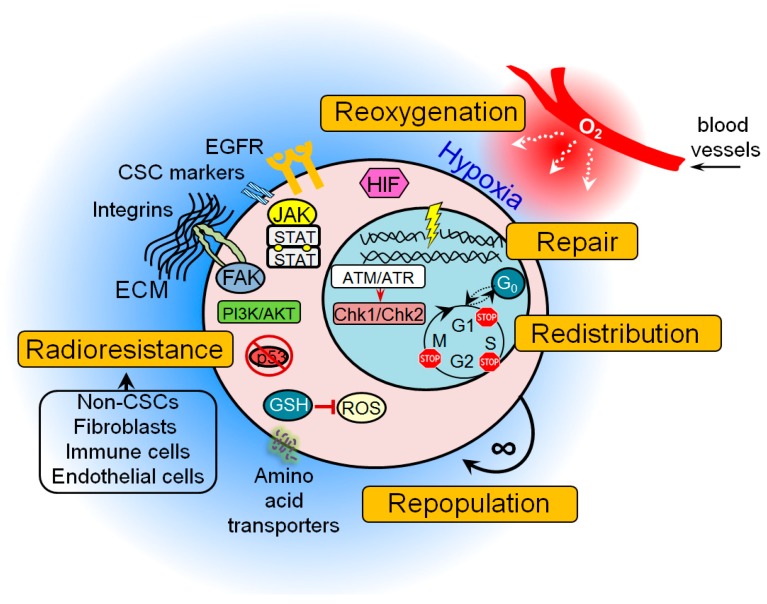
The general mechanisms of cancer stem cell (CSC) radioresistance. Many protective mechanisms are activated in CSCs that result in treatment resistance. These include intrinsic determinants (activation of the pro-survival pathways, enhanced DNA repair capability, protection against oxidative stress, unlimited self-renewal potential, impaired cell cycle arrest, and apoptosis) and extrinsic determinants such as hypoxic microenvironment and a protective CSC niche consisting of the cellular components (e.g., non-CSCs, fibroblasts, immune cells, endothelial cells), soluble factors (e.g., growth factors, hormones and cytokines), and extracellular matrix. ATM—ataxia–telangiectasia mutated; ATR—ATM- and Rad3-Related; Chk1—checkpoint kinase 1; Chk2—checkpoint kinase 2; ECM—extracellular matrix; FAK—focal adhesion kinase; GSH—glutathione; JAK—Janus kinase; EGFR—epidermal growth factor receptor; HIF—hypoxia-inducible factor; PI3K—phosphatidylinositol 3-kinases; ROS—reactive oxygen species.

**Figure 2 cancers-11-00862-f002:**
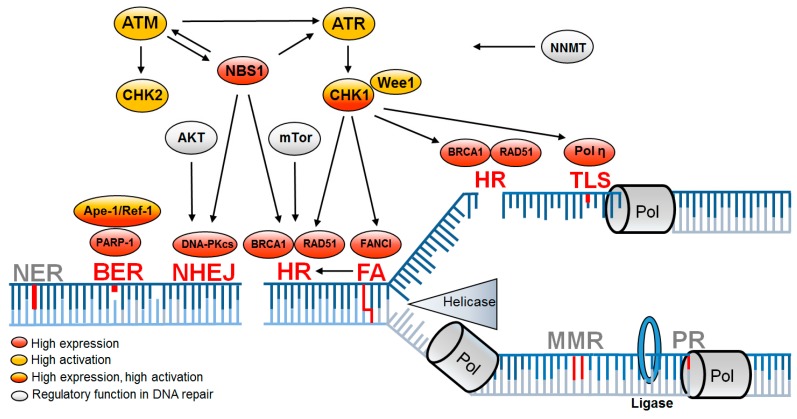
DNA repair pathways altered in CSCs are associated with DNA replication. Replication-associated DNA repair pathways leading to enhanced DNA repair (red) mediate resistance in CSCs, while further replication-associated DNA repair pathways remain unchanged (gray). Upregulation is based on increased expression (red) or increased activation (yellow) or both (red/yellow) and is supported by regulatory proteins (gray). BER—base excision repair; FA—Fanconi anemia; HR—homologous recombination; NER—nucleotide excision repair; NHEJ—non-homologous end joining; MMR—mismatch repair; PR—proof reading; TLS—translesion synthesis [9,68,71,72,73,74,75,76,77,78,79,80,81,82,83,84,85].

**Table 1 cancers-11-00862-t001:** Therapeutic approaches to specifically overcome radiation resistance in cancer stem cells (CSCs). The resistance mechanisms responsible in CSCs after irradiation or chemotherapy and the respective inhibitors used to selectively sensitize them are listed; NSCLC - non-small cell lung cancer; HDAC - histone deacetylases; i-inhibitors.

Drug	Resistance Mechanism	Therapeutic Approach	Tumor	Citation
IR	ATM, ATR, CHK1, PARP-1 upregulation	ATR i + PARP1 i	Glioblastoma	[82]
IR	Replication stress + DDR activation	ATR i + PARP1 i	Glioblastoma	[84]
IR	SSB repair	PARP1 i	Glioblastoma	[160]
IR	DNA-PK activation	DNA-PK i	Glioblastoma	[76]
IR	DNA-PKCS activation	DNA-PKCS i	Glioblastoma	[99]
IR	DDR activation	ATM i	Glioblastoma	[87]
IR	DDR activation	CHK1 i	Glioblastoma	[69]
IR	ZEB1-mediated CHK1 stabilization	ZEB1 depletion by siRNA	Breast	[104]
IR	RAD51 overexpression	RAD51 i	NSCLC	[163]
IR	WEE1 overexpression	WEE1 i	Glioblastoma	[80]
IR	Akt signalling	Akt i	Breast	[166]
IR	mTor signalling	mTorC1/2 inhibition	Glioblastoma	[165]
Cisplatin/paclitaxel	CHK1 activation	CHK1 i	NSCLC	[86]
Radionuclide antibody	CHK1 activation	CHK1 i	Pancreatic	[161]
Irinotecan	CHK1 activation	CHK1 i	Colon	[162]
ICL	ATR/CHK1 activation	ATR depletion by siRNA	Colon	[83]
Olaparib	RAD51 overexpression	RAD51 depletion by shRNA	Breast	[73]
Cisplatin	TLS activity	Pol η depletion by siRNA	Ovary	[79]
Cisplatin	Overexpression of ROS scavengers	ROS scavenger i by 2-ME, 3AT	HNSCC	[170]
Vorinostat	DDR activation and G2 Checkpoint	HDACsi + WEE1 i	Leukemia	[167]
Carboplatin	ALDH1 overexpression	ALDH1 i depletion by siRNA	Ovary	[164]

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
