# Peer review of "Cancer Stem Cells and Radioresistance: DNA Repair and Beyond"

_cancers, 2019, doi:10.3390/cancers11060862_

Round 1

Reviewer 1 Report

This is a nice and comprehensive review to cover the roles and potential mechanisms of cancer stem cells in mediating radioresistance. I recommend to accept the manuscript in its present form for publication.

Author Response

The revised manuscript has incorporated changes based on the reviewer’s suggestions. We are confident that this revision strengthened the manuscript and make it even more relevant for the translational research community.

Reviewer 2 Report

Line 174 should 'pull' be 'pool'?

Line 178 'The self renewal and differentiation capacities...' I am confused by this sentence please revise.

Line 184 This last paragraph is confusing and the grammar slightly clumsy. Please revise.

Line 205 I am not sure that there is a generally accepted view that CSCs maintain genomic stability; there are arguments that CSCs are the source of genomic heterogeneity within the tumour and hence must have genomic instability.

Line 219 Ref 74 makes no mention of increased oxidative damage in CSC and even provides evidence to the contrary. Line 221 'This results in a lower number of DSBs' This statement is confusing and needs revision. The activation of DDR probably results in enhanced repair of DSBs rather than reduced numbers of DSBs.

Line 236 Authors state 'a number of studies', but refer only to a chapter in a textbook which is difficult/impossible to find the full text of, and appears not to be in English. Please list the original research manuscripts which detail this argument, or give less prominence to this line of argument.

Line 262; define 'bulk' cells

Line 284, this paragraph should include discussion of Peter Dirks paper on fate mapping of human glioblastoma Nature 2018.

Line 291 'IR is able to induce / tumour recurrence and metastasis.' Author references a review article here. The original review is much more guarded and in no way states that IR induces tumour recurrence.  It is accepted that IR can promote tumour cell invasion and migration.

Line 305 check greek symbol

Line 336 'exacerbating DNA damage repair' please explain

Line 343 please explain the relation of CSC to 'cancer cells' in general. Are you referring to non stem differentiated cancer cells? The article would benefit from consistent nomenclature in this regard. 'This way some CSCs are spared by radiation' Please supply reference from original manuscript for support of this statement.

Line 353 The authors propose that a big challenge to IR treatment is tumour cell plasticity with conversion of non CSC to CSC following irradiation. Ref 123, 124,  does not support this, ref 125 provides some evidence. The idea of plasticity of non CSC to CSC has been highly controversial. The established dogma would suggest that radiation kills non CSC, thereby leading to apparent enrichment of CSC in an irradiated population (i.e. the enrichment is actually just non-CSC depletion rather than plasticity).  Inclusion of recent nature comms paper by S Niclou might help the author's arguments here.

Line 464 Authors may wish to mention current clinical trials looking at DDR inhibition in GBM and other tumour sites; ATM inhibitor and PARP inhibitor trials are currently recruiting.

Author Response

-           Line 174. should 'pull' be 'pool'?

We have corrected the sentences as follows: “Tumor recurrences arise after radiotherapy depending on the actively proliferating tumor progenitor cells induced by tumor reoxygenation”.

-           Line 178. The self-renewal and differentiation capacities...' I am confused by this sentence please revise.

The sentence was revised as follows: “The unlimited self-renewal capacity of CSCs suggests their role in tumor repopulation between the radiotherapy fractions.”

-           Line 184 This last paragraph is confusing and the grammar slightly clumsy. Please revise.

According to the reviewer comment, the paragraph was revised as follows: “Hereby, the radiobiological concept of 5R’s has to be considered in context of the cancer stem cell tumor model to explain the effect of radiotherapy on tumor tissues. In the next chapters, we discuss the DNA repair and further mechanisms of CSC therapy resistance to identify them as potential targets for the specific elimination of the CSC populations.”

-           Line 205. I am not sure that there is a generally accepted view that CSCs maintain genomic stability; there are arguments that CSCs are the source of genomic heterogeneity within the tumour and hence must have genomic instability.

We absolutely agree with the reviewer and have changed the sentence as follows:

“It has already been observed that CSC, comparable to tissue stem cells, has an altered DNA damage response and repair pathways. In tissue stem cells, the alteration of DNA repair pathways ensures error-free DNA preservation. In CSC, on the other hand, this causes resistance to exogenously induced DNA damage that leads to the failure of tumor therapy. After irradiation, resistance correlates with significantly less DNA damage in glioma and breast cancer stem cells. This also confirms the generally accepted view that tissue stem cells guarantee the maintenance of genomic stability and CSCs ensure the survival of the entire tumor population”.

-           Line 219 Ref 74 makes no mention of increased oxidative damage in CSC and even provides evidence to the contrary. Line 221 'This results in a lower number of DSBs' This statement is confusing and needs revision. The activation of DDR probably results in enhanced repair of DSBs rather than reduced numbers of DSBs.

Line 219: We thank the reviewer for the advice, apologize for the mistake and have now inserted the correct citation, Diehn et al., 2009.

Line 221: This sentence refers to the results described by Diehn et al., 2009 and also observed by Bartkova et al., 2010.

-           Line 236 Authors state 'a number of studies', but refer only to a chapter in a textbook which is difficult/impossible to find the full text of, and appears not to be in English. Please list the original research manuscripts which detail this argument, or give less prominence to this line of argument.

For simplicity purposes we had referred in this review article to an earlier review article (listed in PubMed) which of course was also published in English. To simplify matters, we have now also listed the corresponding studies.

-           Line 262; define 'bulk' cells.

We thank the reviewer for the note and have now replaced the term bulk cells with non-CSC.

-           Line 284, this paragraph should include discussion of Peter Dirks paper on fate mapping of human glioblastoma Nature 2018.

We have included the manuscript of Dirk and colleagues in the discussion as suggested by the reviewer.

-           Line 291 'IR is able to induce / tumour recurrence and metastasis.' Author references a review article here. The original review is much more guarded and in no way states that IR induces tumour recurrence.  It is accepted that IR can promote tumour cell invasion and migration.

We absolutely agree with the reviewer and apologize for this misleading paragraph. We have now revised the paragraph.

-           Line 305 check greek symbol

The Greek symbols were corrected.

-           Line 336 'exacerbating DNA damage repair' please explain

To make this statement more clear, we have replaced the word “exacerbating” by “further impairing”.

-           Line 343 please explain the relation of CSC to 'cancer cells' in general. Are you referring to non stem differentiated cancer cells? The article would benefit from consistent nomenclature in this regard. 'This way some CSCs are spared by radiation' Please supply reference from original manuscript for support of this statement.

According to the reviewer comment, the words “cancer cells” were replaced by “non-CSC” in this sentence and in the entire manuscript where it is appropriate. 

The reference for the study of Diehn et al., 2009 has been added to the above-mentioned paragraph.    

-           Line 353 The authors propose that a big challenge to IR treatment is tumour cell plasticity with conversion of non CSC to CSC following irradiation. Ref 123, 124,  does not support this, ref 125 provides some evidence. The idea of plasticity of non CSC to CSC has been highly controversial. The established dogma would suggest that radiation kills non CSC, thereby leading to apparent enrichment of CSC in an irradiated population (i.e. the enrichment is actually just non-CSC depletion rather than plasticity).  Inclusion of recent nature comms paper by S Niclou might help the author's arguments here.

We appreciate reviewer for this comment. We have added the reference to the study of Niclou and colleagues and removed the references which are less relevant to this topic. The sentence has been revised as follows: “This has been shown for some types of malignancies including glioblastoma and breast cancer and represents a possible challenge to IR treatment“.

-           Line 464 Authors may wish to mention current clinical trials looking at DDR inhibition in GBM and other tumour sites; ATM inhibitor and PARP inhibitor trials are currently recruiting.

We thank the reviewer for this information and have now referred to corresponding clinical studies with the sentence: The use of PARP inhibitors in combination with temozolomide and radiotherapy is currently being tested in clinical trials (Fulton et al., 2017, Leseur et al., 2019).

Reviewer 3 Report

The review is generally well-written, there are however some things which need to be addressed, in particular in relation to the DNA damage response.

Major comments:

1.      Although the authors describe the different CSC models later in the text, in the first section (rows 58-64) only the hierarchical CSC model is described. It would be good to nuance the picture already here by briefly introducing the updated CSC model (for example presented by Batlle & Clevers, Nature Medicine volume 23, pages 1124–1134 (2017)).

2.      Row 83. An accumulation of the unrepairable radiation-induced DSBs results in cell cycle arrest, tumor cell senescence and death. This sentence could be interpreted as all DSBs are unrepairable, please rephrase it since many breaks are properly re-joined.

3.      Row 203: This sentence about an error-free preservation of DNA is in contrast to what was previously stated about an accumulation of DNA mutations in CSC on row 136-137? One reference is from normal stem cells, is this from where this was taken?

4.      Row 205-207: Where is this ”generally accepted view” from? Recent whole genome sequencing data suggest that the mutational burden is high in several cancer types with a more stem-like phenotype. Are there any reports of a maintained genomic stability in CSCs? I have only seen the opposite since quiescent cells (lacking S-phase) would be more prone to use NHEJ.

5.      Row 223-224: It is not correct generally that HR is of outstanding importance, do you mean …repair pathways in CSCs?

6.      Figure 2: The text (and cited references) concern mainly elevated activation/expression of HR components. Still NHEJ and DNA-PKcs are highlighted as having high expression. Where was this information from? I would be helpful to add all references which forms the basis for Figure 2 in the end of the legend, or alternatively, to make sure that they are all included in the text. The title and legend also specify replication-associated DNA repair. NHEJ can also act in S-phase but its inclusion here is perhaps misleading since it mainly is HR which deals with stalled replication forks  (for example described by Kakarougkas A, Jeggo PA, Br J Radiol. 2014 Mar;87(1035):20130685). By regulational (white colour), do you mean regulatory?

7.      There are reports (for example from Maugeri-Saccà M, Mol Cancer Ther August 1 2012 (11) (8) 1627-1636) showing that the DNA repair differences seen in the early papers by Bao et al were due to cell cycle kinetics. There are also more studies which suggest a reduced DNA damage response upon IR in CSC-models. Although a number of reports suggest elevated DNA repair capacity as a CSC phenotype, I would generally suggest to write altered instead of increased DNA damage response, i.e. to modify the title and text in figure 2 legend.

Minor comments:

Row 81: DNA injury is a non-standard term compared DNA damage.

Row 117: I would suggest to add ”the more” before error-prone…… (and ”the” more accurate…), since NHEJ is still correctly repairing many, but not all breaks.

Row 120: please add ”late” before S-phase

The capitalised Rs appear strange in running text (even though they belong to the 5 Rs), one is for example in row 140

Row 146 Do you mean cues instead of clues?

Row 146-8: I would not call the tumour a tissue directly. Tumours are arising within structured tissues but are not organised in the same way. Please rephrase the sentence.

Row 174:The word ”pull” is unclear in this context, please replace it to a more suitable word choice.

Row 186-187: …DNA repair and beyond mechanisms…. Please rephrase.

Row 252: ….nicotinamide N-methyltransferase (NNMT)…

Row 305 and later: The TNF alpha symbol needs to be modified

Table 1: Radionuclide instead of radionucleotide? HDACsi instead of HADACsi?

A few ”the” or plural s are missing, or are not needed in certain parts of the text, please read through to make sure the English is grammatically correct.

Author Response

1.      Although the authors describe the different CSC models later in the text, in the first section (rows 58-64) only the hierarchical CSC model is described. It would be good to nuance the picture already here by briefly introducing the updated CSC model (for example presented by Batlle & Clevers, Nature Medicine volume 23, pages 1124–1134 (2017).

We appreciate reviewer for this comment. We have added the references which provide an insight to the evolved CSC model taking in account CSC plasticity upon genetic changes, epigenetic reprogramming and microenvironmental stimuli.  

2.      Row 83. An accumulation of the unrepairable radiation-induced DSBs results in cell cycle arrest, tumor cell senescence and death. This sentence could be interpreted as all DSBs are unrepairable, please rephrase it since many breaks are properly re-joined.

The text was revised according to this suggestion as follows: “If the levels of radiation-induced DSBs exceed the DNA reparation capacity of tumor cells, this might results in cell cycle arrest, tumor cell senescence and death”.

3.      Row 203: This sentence about an error-free preservation of DNA is in contrast to what was previously stated about an accumulation of DNA mutations in CSC on row 136-137? One reference is from normal stem cells, is this from where this was taken?

We absolutely agree with the reviewer and apologize for this misleading paragraph. We have now revised the paragraph (see also expert 2).

4.      Row 205-207: Where is this ”generally accepted view” from? Recent whole genome sequencing data suggest that the mutational burden is high in several cancer types with a more stem-like phenotype. Are there any reports of a maintained genomic stability in CSCs? I have only seen the opposite since quiescent cells (lacking S-phase) would be more prone to use NHEJ.

We absolutely agree with the reviewer and apologize for this misleading paragraph. We have now revised the paragraph (see also expert 2).

We agree with the reviewer that cell cycle effects also have a particular role in the resistance of tumor stem cells and have tried to discuss this in Line 255. We have also included the citation mentioned

5.      Row 223-224: It is not correct generally that HR is of outstanding importance, do you mean …repair pathways in CSCs?

We thank the reviewer for this hint and have now referred the sentence to the role of DNA repair by homologous recombination, which is exclusively emphasized in CSCs.

6.      Figure 2: The text (and cited references) concern mainly elevated activation/expression of HR components. Still NHEJ and DNA-PKcs are highlighted as having high expression. Where was this information from? I would be helpful to add all references which forms the basis for Figure 2 in the end of the legend, or alternatively, to make sure that they are all included in the text. The title and legend also specify replication-associated DNA repair. NHEJ can also act in S-phase but its inclusion here is perhaps misleading since it mainly is HR which deals with stalled replication forks  (for example described by Kakarougkas A, Jeggo PA, Br J Radiol. 2014 Mar;87(1035):20130685). By regulational (white colour), do you mean regulatory?

We agree with the reviewer that mentioning the citations on which the illustration is based would be helpful for the reader and have now added this at the end of the legend. The work of Kakarougkas et al., 2014 has now been added to the general introduction, along with other review articles on DNA repair.

We agree with the reviewer that the significance of the NHEJ in the S phase is of lesser importance. However, there is a publication that highlights the importance of NHEJ on stopped replication forks and wanted to include this in this article.

7.      There are reports (for example from Maugeri-Saccà M, Mol Cancer Ther August 1 2012 (11) (8) 1627-1636) showing that the DNA repair differences seen in the early papers by Bao et al were due to cell cycle kinetics. There are also more studies which suggest a reduced DNA damage response upon IR in CSC-models. Although a number of reports suggest elevated DNA repair capacity as a CSC phenotype, I would generally suggest to write altered instead of increased DNA damage response, i.e. to modify the title and text in figure 2 legend.

We agree with the reviewer that there are also studies in which a reduction of DNA repair mechanisms has been observed. We had only quoted one review article so far, but have now included the original citations (see line 236 and the commentary by expert 2). We have also included the good suggestion of the reviewer and have replaced up-regulated by altered.

Minor comments

Row 81: DNA injury is a non-standard term compared DNA damage.

We have replaced the term DNA injury by DNA damage.

Row 117: I would suggest to add ”the more” before error-prone…… (and ”the” more accurate…), since NHEJ is still correctly repairing many, but not all breaks.

We agree with the reviewer and have now modified the sentence according to the reviewer's suggestion.

Row 120: please add ”late” before S-phase

We have now renamed S-Phase to late S-Phase.

The capitalised Rs appear strange in running text (even though they belong to the 5 Rs), one is for example in row 140

We've removed the capitalized Rs.

Row 146 Do you mean cues instead of clues?

We have now written cues instead of clues, as suggested by the reviewer.

Row 146-8: I would not call the tumour a tissue directly. Tumours are arising within structured tissues but are not organised in the same way. Please rephrase the sentence.

We have rewritten the sentence as follows: Tumors show growth comparable to that of rapidly proliferating tissues with a comparably high demand for oxygen and nutrients from the blood vessels" and hope the reviewer agrees with this improvement.

Row 174:The word ”pull” is unclear in this context, please replace it to a more suitable word choice.

We agree with the reviewer and have now adapted the sentence as follows: "Tumor recurrences arise after radiotherapy depending on the actively proliferating tumor progenitor cells induced by reoxygenation."

Row 186-187: …DNA repair and beyond mechanisms…. Please rephrase.

We thank the reviewer for the constructive hint and have improved the sentence according to the suggestion. It now reads: "In the next chapters we discuss DNA repair and further mechanisms of CSC therapy resistance to identify them as potential targets for the specific elimination of CSC populations".

Row 252: ….nicotinamide N-methyltransferase (NNMT)…

We would like to thank the reviewer for this information and have inserted nicotinamide into the text.

Row 305 and later: The TNF alpha symbol needs to be modified

The Greek symbols were corrected.

Table 1: Radionuclide instead of radionucleotide? HDACsi instead of HADACsi?

We thank the reviewer and have corrected the noted errors in the table.

A few ”the” or plural s are missing, or are not needed in certain parts of the text, please read through to make sure the English is grammatically correct.

The manuscript has been carefully proofread and spell checked. 

Round 2

Reviewer 3 Report

Please correct the spelling of integrins (now intergins) in Figure 1.

Row 244: Should it be both? increased up-regulation

Author Response

 Reply to the reviewers' comments:

1.            Please correct the spelling of integrins (now intergins) in Figure 1.

-     The spelling was corrected and updated figure was included in the manuscript.

2.            Row 244: Should it be both? increased up-regulation

-     We have corrected the sentences as follows: “Glioblastoma cell lines show an up-regulation of HR genes”